# Bioactive Products from Endophytic Fungi of Sages (*Salvia* spp.)

**Beata Zimowska [1], Monika Bielecka [2,\*], Barbara Abramczyk [3] and Rosario Nicoletti [4,5]**

[1]  Department of Plant Protection, University of Life Sciences, Leszczyńskiego 7, 20-069 Lublin, Poland; beata.zimowska@up.lublin.pl

[2]  Department of Pharmaceutical Biotechnology, Wroclaw Medical University, Borowska 211A, 50-556 Wroclaw, Poland

[3]  Department of Agricultural Microbiology, Institute of Soil Science and Plant Cultivation, Czartoryskich 8, 24-100 Puławy, Poland; babramczyk@iung.pulawy.pl

[4]  Council for Agricultural Research and Economics, Research Centre for Olive, Fruit and Citrus Crops, 81100 Caserta, Italy; rosario.nicoletti@crea.gov.it

[5]  Department of Agricultural Sciences, University of Naples 'Federico II', 80055 Portici, Italy

\*  Correspondence: monika.bielecka@umed.wroc.pl

**Abstract:** In the aim of implementing new technologies, sustainable solutions and disruptive innovation to sustain biodiversity and reduce environmental pollution, there is a growing interest by researchers all over the world in bioprospecting endophytic microbial communities as an alternative source of bioactive compounds to be used for industrial applications. Medicinal plants represent a considerable source of endophytic fungi of outstanding importance, which highlights the opportunity of identifying and screening endophytes associated with this unique group of plants, widespread in diverse locations and biotopes, in view of assessing their biotechnological potential. As the first contribution of a series of papers dedicated to the Lamiaceae, this article reviews the occurrence and properties of endophytic fungi associated with sages (*Salvia* spp.).

**Keywords:** sage; endophytes; bioprospecting; bioactive compounds; medicinal plants; Lamiaceae

## 1. Introduction

Endophytic fungi are defined as fungi inhabiting tissues and organs of healthy plants during certain stages of their life cycle without causing apparent symptoms. The concept of endophytism, introduced by De Bary in 1866 and almost completely neglected for over a century, has recently become of common usage concomitantly to advances in knowledge on occurrence and functions of this component of biodiversity. Increasing attention by the scientific community is boosted by the opportunity to exploit the unique aptitudes and properties of these microbial associates of plants [1].

As a consequence of the long-term association of endophytes with medicinal plants, based on mutually beneficial relationships, the former may also participate in metabolic paths and boost their own natural biosynthetic activity, or may gain some genetic information to synthesize biologically active compounds closely related to those directly produced by the host plant [2,3]. Endophytic fungi derived from medicinal plants are becoming more and more popular, due to specific modes of action and the ability to provide multiple benefits, which make them relevant for both agricultural and pharmaceutical applications [3]. This review is devoted to an analysis of the biochemical potential of endophytic fungi reported from species of sage (*Salvia* spp.), examining the advances in this particular field made by the scientific community in recent years.

## 2. *Salvia*: The Largest Genus of Lamiaceae

Lamiaceae is one of the most important herbal families, including a wide variety of plants with multiple medical, culinary and industrial applications. Within the subfamily Nepetoideae, sage species are ascribed to the genus *Salvia*, a name deriving from the Latin word "*salvere*", which refers to the curative properties of these plants. It represents the largest genus of the family counting between 700 and 1000 species [4]. Uncertainty of this number is basically due to the broad geographical range of distribution, covering all continents and climatic areas, which makes taxonomic verification problematic.

As for many plants and other organisms, the application of biomolecular techniques in taxonomy has determined several basic reassessments in classification. Until a few years ago, the genus name *Salvia* was only used for species displaying the typical morphological features of sage. Nevertheless, recent systematic work has emphasized close relationships with the genera *Dorystaechas*, *Meriandra*, *Perovskia*, *Rosmarinus* and *Zhumeria*, which resulted to be clearly embedded in *Salvia* in dedicated phylogenetic analyses, so that their separation is no more justified [4,5]. Although not consolidated in the common use yet, this new taxonomic sorting is basically followed in this paper. However, by reason of several peculiar aspects concerning geographical distribution and biotechnological applications, the species *Salvia rosmarinus* (=*Rosmarinus officinalis*) will be the subject of a dedicated analysis in a forthcoming paper.

Medicinal properties of sages derive from their ability to produce a multitude of bioactive secondary metabolites, many of which have been reported for antibiotic, antitumor, antiviral, antiprotozoal, insecticidal and antioxidant effects, or even to be responsible for allelopathic interactions with other plants [6]. These varied bioactivities are reflected by quite diverse chemical structures. In fact, besides flavonoids and simple phenolic compounds like caffeic, rosmarinic and salvianolic acids, which are mainly known for their radical scavenging effects, these products include monoterpenoids, sesquiterpenoids, triterpenoids and diterpenoids. Structural diversity is particularly evident within this latter grouping, including labdanes, ent-kauranes, abietanes, icetexanes, clerodanes, and pimaranes, as well as phenolic diterpenoids, such as carnosol and carnosic acid [6]. Moreover, some abietanes are rearranged, to form the important scaffold of tanshinones [7]. These latter products are particularly considered for pharmaceutical application based on their antioxidant [8], antibacterial [9], antidiabetic [10], anti-inflammatory [11], and antiproliferative [12] properties, and are currently the subject of a specific project at our laboratories, in which the species *Salvia abrotanoides* (formerly *Perovskia abrotanoides*) and *Salvia yangii* (formerly *Perovskia atriplicifolia*), regarded as an alternative source of tanshinones, are analyzed through combined metabolomics and transcriptomics approaches, also with reference to the associated endophytic fungi.

## 3. Ecology and Occurrence

As introduced above, endophytic fungi are polyphyletic groups of microorganisms, which asymptomatically colonize healthy tissues of different parts of living plants such as stems, leaves or roots. Their diversity is huge, and it has been estimated that every plant hosts several endophytic species, among which at least one shows host specificity [13,14]. Through the evolutionary processes, endophytic fungi have developed different symbiotic relationships with their host plants [3]. Furthermore, many species are reported to exhibit multiple ecological roles as both endophytes and pathogens. However, it is not clear whether the same genotypes can play both these roles with equal success. Understanding the mechanisms responsible for the conversion between so different outcomes of the ecological interaction represents one among many frontiers in endophyte biology [15,16]. One of the mechanisms developed by plants during the long-term co-evolution with microbial associates is the ability to produce antibiotic compounds. Simultaneously, many endophytes have developed an important transformative capacity and/or tolerance to these products which in a large part determines the colonization range of their hosts [17]. In turn, endophytes can influence growth and development of host plants, and enhance their resistance to biotic and abiotic stresses by releasing bioactive

metabolites [18], to such an extent that in natural habitats some plant species require to be supported by endophytic fungi for stress tolerance and survival [19].

The diversity of endophytic fungi associated with medicinal plants is largely affected by ecological or environmental factors. Particularly, temperature, humidity and soil nutritional conditions influence the quality and quantity of secondary metabolites synthesized by the hosts, which in turn affect the population structure of endophytic fungi. The species composition of endophyte communities also differs in organ and tissue specificities, as a result of their adaptation to different physiological conditions in hosts [3,20].

The analysis of the recent literature shows that species of sage (*Salvia* spp.) host diverse communities of fungal endophytes. As many as 64 different taxa belonging to 38 genera, with a clear prevalence of Ascomycetes, have been reported so far (Table 1). Most observations concern the species *Salvia miltiorrhiza* and *S. abrotanoides*, respectively, with 28 and 24 records. There is an evident correspondence between the *Salvia* species and the geographical area. In fact, all isolations concerning *S. miltiorrhiza* come from several provinces of China, while the available findings from *S. aegyptiaca* come from Egypt, and those referring to *S. abrotanoides* derive from an Iranian study and from the activity currently in progress at our laboratories. Despite the fact that isolations have been carried out from any plant organ (Figures 1 and 2), no indications concerning a specific association with roots or the aerial parts can be inferred. The access to biomolecular methods as a taxonomic tool has generally enabled to perform reliable identification at the species level, with the exception of a Chinese study concerning seeds of *S. miltiorrhiza*, where sorting was basically limited to the class level despite the wide variation observed [21]. The repeated findings in several species/locations mostly refer to strains provisionally identified at the genus level, particularly *Alternaria*, which seems to be of quite common occurrence on sages regardless to the plant part used for isolations. At the species level, there are only two cases with more than just one record—that is, *Chaetomium globosum* and *Didymella* (=*Phoma*) *glomerata*, both from *S. miltiorrhiza* at two different locations in China. However, the recovery of these species from both roots and leaves may represent a possible indication of a more regular association with this plant, which should be taken into consideration in further studies.

**Table 1.** Endophytic fungi reported from *Salvia* spp.

| Endophyte [1] | Plant Species/Organ | Location, Country | Reference |
|---|---|---|---|
| *Acremonium sclerotigenum* | *S. abrotanoides*/root | Zoshk, Iran | [22] |
| *Alternaria alternata* | *S. miltiorrhiza*/flower | Shandong, China | [23] |
| | *S. aegyptiaca*/leaf | Gebel Elba, Egypt | [24] |
| *Alternaria chlamydosporigena* | *S. abrotanoides*/root | Zoshk, Iran | [22] |
| *Alternaria* sp. | *S. miltiorrhiza*/root | Beijing, China | [25] |
| | *S. miltiorrhiza*/seed | Northwest China | [21] |
| | *S. miltiorrhiza*/root, shoot, leaf | Henan, China | [26] |
| | *S. abrotanoides*/leaf, stem | Wroclaw, Poland | this paper |
| | *S. yangii*/leaf, stem | Wroclaw, Poland | this paper |
| *Alternaria tenuissima* | *S. przewalskii*/root | Longxi, China | [27] |
| *Aspergillus brasiliensis* | *S. aegyptiaca*/leaf | Gebel Elba, Egypt | [24] |
| *Aspergillus foeniculicola* | *S. miltiorrhiza*/root | Shaanxi, China | [28] |
| *Aspergillus nidulans* | *S. aegyptiaca*/leaf | Gebel Elba, Egypt | [24] |
| *Aspergillus niger* | *S. aegyptiaca*/leaf | Gebel Elba, Egypt | [24] |
| *Aspergillus* sp. | *S. miltiorrhiza*/root | Beijing, China | [25] |
| | *S. abrotanoides*/leaf | Zoshk, Iran | [22] |
| *Aspergillus terreus* | *S. aegyptiaca*/leaf | Gebel Elba, Egypt | [24] |
| *Aureobasidium* sp. | *S. miltiorrhiza*/seed | Northwest China | [21] |
| *Cadophora* sp. | *S. miltiorrhiza*/root | Beijing, China | [25] |
| *Canariomyces microsporus* | *S. abrotanoides*/leaf | Zoshk, Iran | [22] |
| *Cephalosporium acremonium* | *S. aegyptiaca*/leaf | Gebel Elba, Egypt | [24] |
| *Chaetomium globosum* | *S. miltiorrhiza*/root | Shanluo, China | [29] |
| | *S. miltiorrhiza*/'aerial part' | Shenyang, China | [30] |

**Table 1.** *Cont.*

| Endophyte [1] | Plant Species/Organ | Location, Country | Reference |
|---|---|---|---|
| *Chaetomium* sp. | *S. officinalis*/stem | Beni-Mellal, Morocco | [31] |
| | | Giza, Egypt | [32] |
| *Cladosporium cladosporioides* | *S. aegyptiaca*/leaf | Gebel Elba, Egypt | [24] |
| *Clonostachys rosea* | *S. miltiorrhiza*/root | Beijing, China | [25] |
| *Colletotrichum gloeosporioides* | *S. miltiorrhiza*/'aerial part' | Shenyang, China | [33] |
| *Colletotrichum* sp. | *S. aegyptiaca*/leaf | Gebel Elba, Egypt | [24] |
| | *S. yangii*/leaf | Wroclaw, Poland | this paper |
| *Coniolariella hispanica* | *S. abrotanoides*/root | Kalat, Iran | [22] |
| *Curvularia papendorfii* | *S. aegyptiaca*/leaf | Gebel Elba, Egypt | [24] |
| *Diaporthe* sp. | *S. miltiorrhiza*/stem | Sichuan, China | [34] |
| | *S. abrotanoides*/stem | Wroclaw, Poland | this paper |
| | *S. yangii*/stem | Wroclaw, Poland | this paper |
| *Didymella glomerata* | *S. miltiorrhiza*/root | Beijing, China | [25] |
| | *S. miltiorrhiza*/leaf | Shangluo, China | [35] |
| *Didymella pedeiae* | *S. miltiorrhiza*/root | Beijing, China | [25] |
| *Filobasidium* sp. | *S. miltiorrhiza*/seed | Northwest China | [21] |
| *Fusarium dlaminii* | *S. abrotanoides*/root | Darrud, Iran | [22] |
| *Fusarium oxysporum* | *S. aegyptiaca*/leaf | Gebel Elba, Egypt | [24] |
| *Fusarium proliferatum* | *S. miltiorrhiza*/root | Shandong, China | [23] |
| *Fusarium redolens* | *S. miltiorrhiza*/root | Beijing, China | [25] |
| *Fusarium* sp. | *S. miltiorrhiza*/root | Beijing, China | [25] |
| | *S. abrotanoides*/root, stem | Wroclaw, Poland | this paper |
| | *S. yangii*/root, stem | Wroclaw, Poland | this paper |
| *Juxtiphoma eupyrena* | *S. miltiorrhiza*/root | Beijing, China | [25] |
| *Leptosphaeria* sp. | *S. miltiorrhiza*/root | Beijing, China | [25] |
| *Neocosmospora solani* | *S. abrotanoides*/root | Kalat, Iran | [22] |
| *Niesslia ligustica* | *S. abrotanoides*/root | Darrud, Iran | [22] |
| *Paecilomyces* sp. | *S. miltiorrhiza*/root | Beijing, China | [36] |
| *Paraphoma radicina* | *S. abrotanoides*/root | Zoshk, Iran | [22] |
| *Penicillium canescens* | *S. abrotanoides*/root | Zoshk and Kalat, Iran | [22] |
| *Penicillium charlesii* | *S. abrotanoides*/root | Zoshk and Kalat, Iran | [22] |
| *Penicillium chrysogenum* | *S. abrotanoides*/root | Zoshk, Iran | [22] |
| *Penicillium citrinum* | *S. aegyptiaca*/leaf | Gebel Elba, Egypt | [24] |
| *Penicillium commune* | *S. aegyptiaca*/leaf | Gebel Elba, Egypt | [24] |
| *Penicillium murcianum* | *S. abrotanoides*/root | Kalat, Iran | [22] |
| *Penicillium* sp. | *S. abrotanoides*/root | Zoshk and Kalat, Iran | [22] |
| *Pestalotiopsis mangiferae* | *S. aegyptiaca*/leaf | Gebel Elba, Egypt | [24] |
| *Petriella setifera* | *S. miltiorrhiza*/root | Beijing, China | [25] |
| *Phaeoacremonium rubrigenum* | *S. abrotanoides*/root | Zoshk, Iran | [22] |
| *Phoma herbarum* | *S. miltiorrhiza*/seed | China | [37] |
| *Psathyrella candolleana* | *S. abrotanoides*/root | Zoshk, Iran | [22] |
| *Purpureocillium lilacinum* | *S. abrotanoides*/root | Darrud, Iran | [22] |
| *Sarocladium kiliense* | *S. miltiorrhiza*/root | Beijing, China | [25] |
| *Schizophyllum commune* | *S. miltiorrhiza*/root | Shandong, China | [23] |
| *Simplicillium cylindrosporum* | *S. abrotanoides*/root | Darrud, Iran | [22] |
| *Talaromyces pinophilus* | *S. miltiorrhiza*/'aerial part' | Shenyang, China | [38] |
| *Talaromyces* sp. | *S. abrotanoides*/root | Zoshk and Kalat, Iran | [22] |
| *Talaromyces verruculosus* | *S. abrotanoides*/root | Zoshk and Kalat, Iran | [22] |
| *Trametes hirsuta* | *S. miltiorrhiza*/root | Shandong, China | [23] |
| *Trichocladium griseum* | *S. aegyptiaca*/leaf | Gebel Elba, Egypt | [24] |
| *Trichoderma asperellum* | *S. abrotanoides*/root | Zoshk, Iran | [22] |
| *Trichoderma atroviride* | *S. miltiorrhiza*/root | Shangluo, China | [39] |
| *Trichoderma hamatum* | *S. officinalis*/root | Bonn, Germany | [40] |
| *Trichoderma viride* | *S. aegyptiaca*/leaf | Gebel Elba, Egypt | [24] |
| *Xylomelasma* sp. | *S. miltiorrhiza*/root | Beijing, China | [25] |

[1] Species are reported according to the latest accepted name, which might not be the same as the one used in the corresponding reference.

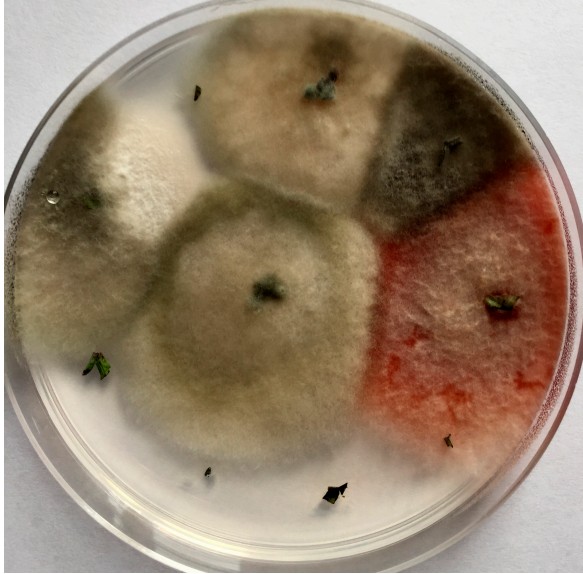

**Figure 1.** Isolation of endophytic fungi from leaf of *Salvia yangii* (original from work currently in progress at our laboratories).

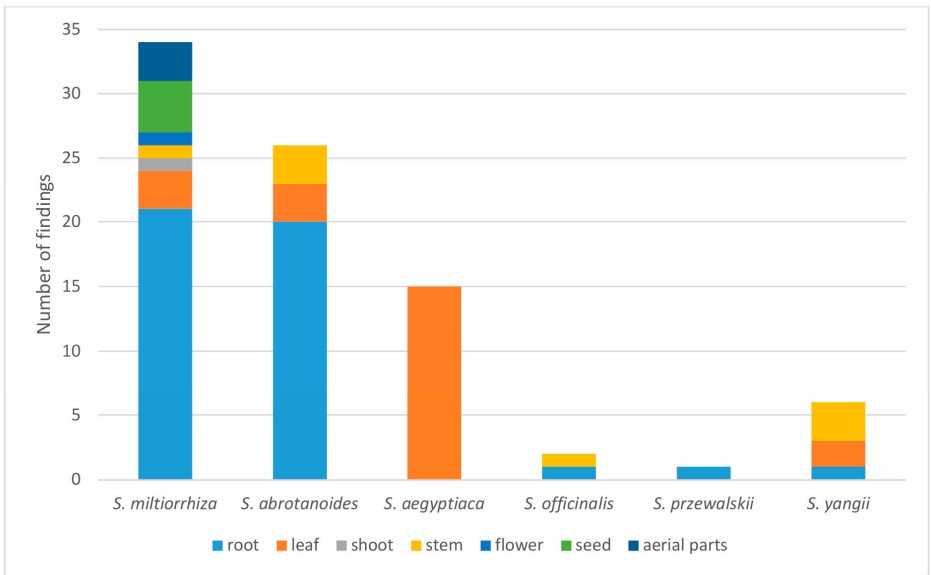

**Figure 2.** Graphic representation of findings concerning endophytic fungi of *Salvia* spp. based on data reported in Table 1.

## 4. Biochemical Properties

A large part of literature on the occurrence of endophytic fungi of *Salvia* spp. deals with their ability to produce bioactive compounds (Figure 3), focusing on structure elucidation and possible applications. Some studies have been limited to a partial characterization of culture filtrates or their extracts, highlighting general antibacterial, antioxidant or antifungal properties [23–25,41], while in other cases the basic constituents have been identified and extracted for assessments concerning their bioactivity. An annotated list of these products is reported in Table 2.

**Table 2.** Bioactive secondary metabolites produced by endophytic fungi from *Salvia* spp.

| Secondary Metabolite | Producing Species/Strain | Bioactivity | Reference |
|---|---|---|---|
| N-Acetylanthranilic acid | *Penicillium* sp. <br> *Talaromyces* sp. | | [22] |
| Altenuene | *Alternaria* sp./Samif01 <br> *Alternaria tenuissima*/SP-07 | | [42] <br> [27] |
| 2-*epi*-Altenuene | *Alternaria* sp./Samif01 | | [42] |
| 2-Acetoxy-2-*epi*-altenuene | *Alternaria* sp./Samif01 | | [42] |
| 3-*epi*-Dihydroaltenuene A | *Alternaria* sp./Samif01 | Radical scavenging | [42] |
| Altenuisol | *Alternaria* sp./Samif01 | Antibacterial, radical scavenging | [42] |
| Alternariol | *Alternaria* sp./Samif01 <br> *Alternaria tenuissima*/SP-07 | Antibacterial | [42] <br> [27] |
| Alternariol-9-methyl ether | *Alternaria* sp./Samif01 <br> *Alternaria tenuissima*/SP-07 | Antibacterial, antifungal, antinematodal | [43] <br> [27] |
| 4-Hydroxyalternariol-9-methyl ether | *Alternaria* sp./Samif01 | Antibacterial, radical scavenging | [42] |
| Aureonitols A–B | *Chaetomium globosum*/XL-1198 | | [30] |
| Azelaic acid | *Penicillium canescens* <br> *Penicillium charlesii* <br> *Penicillium* sp. <br> *Talaromyces* sp. <br> *Talaromyces verruculosus* | | [22] |
| Caffeic acid | *Paraphoma radicina* <br> *Talaromyces* sp. <br> *Talaromyces verruculosus* | | [22] |
| Chaetoglobosins E–F | *Chaetomium globosum*/XL-1198 | | [30] |
| Chaetomin | *Chaetomium* sp. | Cytotoxic (L5178Y mouse lymphoma) | [32] |
| Chaetomugilin I | *Chaetomium globosum*/XL-1198 | | [30] |
| Chaetoquadrin D | *Xylomelasma* sp./Samif07 | | [44] |
| Chaetoviridin | *Chaetomium globosum*/XL-1198 | | [30] |
| Cochliodinol, isocochliodinol, hydroperoxycochliodinol | *Chaetomium* sp. | Cytotoxic (L5178Y mouse lymphoma) | [31,32] |
| Colletotricholides A–B | *Colletotrichum gloeosporioides*/XL1200 | | [33] |
| Cryptotanshinone | *Coniolariella hispanica* <br> *Paraphoma radicina* <br> *Penicillium canescens* <br> *Penicillium murcianum* | | [22] |
| Daidzein | *Fusarium dlaminii* <br> *Neocosmospora solani* <br> *Paraphoma radicina* <br> *Penicillium canescens* | | [22] |
| Diaporthin | *Xylomelasma* sp./Samif07 | Antibacterial, radical scavenging | [44] |
| 2,6-Dimethyl-5-methoxyl-7-hydroxylchromone | *Xylomelasma* sp./Samif07 | Antibacterial | [44] |
| Equisetin | *Chaetomium globosum*/XL-1198 | Antibacterial, antifungal | [30] |
| Ferruginol | *Trichoderma atroviride* D16 | | [39] |
| Glycitein | *Talaromyces* sp. | | [22] |
| Griseofulvin | *Talaromyces* sp. | | [22] |
| 8-Hydroxy-6-methoxy-3-methylisocoumarin | *Xylomelasma* sp./Samif07 | Antibacterial | [44] |
| 6-Hydroxymethyleugenin, 6-methoxymethyleugenin | *Xylomelasma* sp./Samif07 | Antibacterial | [44] |
| Indole-3-acetic acid | *Penicillium canescens* <br> *Phoma herbarum* D603 <br> *Talaromyces* sp. | | [22] <br> [37] <br> [22] |
| Indole-3-carboxylic acid, 3-formylindole | *Chaetomium* sp. | | [32] |
| Isoeugenitol | *Xylomelasma* sp./Samif07 | Antibacterial, antimycobacterial | [44] |
| Mandelic acid | *Paraphoma radicina* <br> *Talaromyces* sp. | | [22] |
| 6-Methoxymellein | *Xylomelasma* sp./Samif07 | | [44] |
| Nipecotic acid | *Penicillium canescens* | | [22] |
| Paracetamol (acetaminophen) | *Penicillium chrysogenum* <br> *Penicillium* sp. | | [22] |
| Pinophicin A | *Talaromyces pinophilus* | Antibacterial | [38] |
| Pinophol A | *Talaromyces pinophilus* | Antibacterial | [38] |
| Salvianolic acid C | *Didymella glomerata*/D-14 | | [35] |
| Solanapyrones A-C | *Alternaria tenuissima*/SP-07 | Antibacterial | [27] |
| Solanapyrones P-R | *Alternaria tenuissima*/SP-07 | Antibacterial | [27] |
| Solanidine | *Talaromyces* sp. | | [22] |
| Stachydrine | *Fusarium dlaminii* | | [22] |
| Tanshinone I | *Trichoderma atroviride* D16 | | [39] |
| Tanshinone IIA | *Aspergillus foeniculicola*/TR21 <br> *Trichoderma atroviride* D16 | | [28] <br> [39] |
| Trigonelline | *Talaromyces* sp. | | [22] |

Underlined compounds were first characterized from these sources.

**Figure 3.** Chemical structure of some bioactive products from endophytic fungi of *Salvia* spp.

　　Confirming the assumption that endophytic fungi represent a goldmine of chemodiversity [2,45], 12 novel products were obtained from strains associated with *Salvia*. The list includes a fusicoccane diterpene pinophicin A and a polyene pinophol A from *Talaromyces pinophilus* [38]; colletotricholides A-B, two unusual eremophilane acetophenone conjugates from *Colletotrichum gloeosporioides* which are synthesized through a hybrid pathway involving polyketide and sesquiterpene synthase [33]. The novel 2,6-dimethyl-5-methoxyl-7-hydroxylchromone from *Xylomelasma* sp. displayed antibacterial activity, along with a few related eugenin derivatives and isocoumarins [44]. Moreover, there are several novel analogues of products known from *Alternaria*, such as 2-acetoxy-2-*epi*-altenuene and solanapyrones P-R [27,42], and *Chaetomium*, such as hydroperoxycochliodinol [32] and aureonitols

A-B [30]. The latter are structurally related to aureonitol, a known antiviral tetrahydrofuran [46]. As for cochliodinol derivatives, their bioactivity was found to be affected by the position of prenyl substituents in the indole ring systems, while the increased cytotoxicity of hydroperoxycochliodinol is related to the hydroperoxyl function [32].

Strains of *Alternaria* and *Chaetomium* also yielded products, such as alternariols, altenuisol, chaetoviridin and chaetoglobosins, whose antibiotic, antifungal, antiproliferative and radical scavenging properties have been previously described, and reviewed in recent papers [47,48]. More known products previously reported from other fungi are the tetramic acid derivative equisetin and the isocoumarin diaporthin, originally described as phytotoxins [49,50], and griseofulvin, a compound which has found application in dermatology and displayed interesting antitumor properties [51]. Other secondary metabolites which are used as pharmaceuticals are paracetamol, nipecotic acid, mandelic acid and azelaic acid. The latter has displayed antibiotic properties and antiproliferative effect on malignant melanocytes, and is commonly used in dermatology as an antiacne [52]. Moreover, in plants it is reported to be involved in the defense response against disease agents [53]. Other products to be mentioned are the plant hormone indole-3-acetic acid (IAA) and a couple of analogue auxins, which are considered key intermediates in the mutualistic relationship between endophytes and their host plants [2,16].

However, probably the most interesting products of endophytic fungi of *Salvia* species are a series of compounds previously identified as plant metabolites, which are treated in further detail in the next chapter.

## 5. Biotechnological Implications

Long-lasting evolutionary processes taking place together with their host plants have allowed endophytic fungi to work out various strategies enabling them to keep an equilibrium between virulence and plant defense in order to share common habitat. Endophytic fungi not only are able to influence plants' metabolism and physiology by producing unique and specific secondary metabolites, they were also found to produce bioactive natural products originally known exclusively from their host plants, and have elaborated strategies for detoxification by exploiting their biotransformation abilities. These properties make endophytes a perfect target for various biotechnological approaches and further commercial exploitation.

### 5.1. Endophytic Fungi as In Vitro Production Platforms for Plant Secondary Metabolites

The ever-increasing demand for bioactive natural compounds cannot be met at the desired levels by just relying on their extraction from plants, considering that in most instances they are produced at a specific developmental stage or under specific environmental condition, stress, or nutrient availability [54]. Medicinal plants from the *Salvia* genus are often shrubs, thus they may need several years to attain a suitable growth phase for bioactive product accumulation and extraction. Moreover, harvesting medicinally important plants from the wild makes them critically endangered and affects the environmental biodiversity [39]. As for crop plants, although cultivated in a large scale, they often produce the desired metabolites in a low yield, making the production unprofitable. Considering the limitations associated with productivity and vulnerability of plants, fungal endophytes may serve as a renewable and inexhaustible source of bioactive compounds. Many endophytes have experienced long-term symbiotic relationships with their host plants, and through long-term coexistence and direct contact, they have exchanged genetic material [17]. Horizontal gene transfer (HGT), an important evolutionary mechanism observed in prokaryotes, is also thought to be the phenomenon responsible for transmission of genetic material across phylogenetically distant species [55]. As an increasing number of reports indicate a physical clustering of genes for specialized metabolic pathways in plant genomes [56], the HGT phenomenon is believed to be responsible for rapid transfer of whole gene clusters from host plants, conferring "novel traits" to the associated fungi. As a consequence, many endophytic fungi have developed the ability to produce bioactive substances originally known from

their hosts, thus raising the prospect of using such organisms as alternative and sustainable sources. HGT has been proposed to explain the production of tanshinone I, tanshinone IIA and their precursor ferruginol by *Trichoderma atroviride* D16, an endophytic fungus in *S. miltiorrhiza* [39].

Daidzein and glycitein are naturally occurring compounds found in soybeans and other legumes which are produced in plants through the phenylpropanoid pathway and structurally belonging to a class of compounds known as isoflavones. Daidzein is a phytoestrogen with possible pharmaceutical application as menopausal relief, osteoporosis, blood cholesterol lowering, and it is thought to reduce the risk of some hormone-related cancers and heart disease [57], while glycitein has a weaker estrogenic activity [58]. They both were found to be produced by endophytic fungi of *S. abrotanoides*, that is *Penicillum canescens* for daidzein and *Talaromyces* sp. for glycitein [22]. The latter is also able to synthesize trigonelline, an alkaloid originally extracted from *Trigonella foenum-graecum*, known for its antidiabetic properties [59] as well as solanidine, a potato alkaloid. Stachydrine, another alkaloid known from *Medicago sativa*, was found to be synthetized by a strain of *Fusarium dlaminii* inhabiting *S. abrotanoides* [22].

Danshen, dried roots and rhizomes of *S. miltiorrhiza*, is a well-known traditional Chinese herbal medicine [60]. It contains two kinds of bioactive compounds: tanshinones and hydrophilic phenolic acids, the latter being represented by rosmarinic acid, salvianolic acids B-C, and others. Salvianolic acids are mainly responsible for the favorable activities on cardiovascular and cerebrovascular diseases of danshen [61]. Salvianolic acid C was found in both mycelium and fermentation broth of strain D14 of *D. glomerata* in very low yields [35]. This indicates the opportunity to optimize fermentation conditions for achieving its efficient production, or alternatively to enhance its production via regulating the key enzymes involved in the biosynthetic pathway.

Caffeic acid was found in the metabolome profiles of isolates of *Talaromyces* and *Paraphoma* endophytic in *S. abrotanoides* [22]. Besides rosmarinic acid and salvianolic acid B, it is regarded as the major phenolic acid in *S. miltiorrhiza* [62]. A series of caffeic acid derivatives, obtained from *Salvia officinalis* [63,64], showed pronounced leishmanicidal activity, as well as immunomodulatory effects on macrophage functions [65]. Moreover, antibacterial, antifungal and modulatory effects of caffeic acid have been shown in recent studies [66].

Tanshinones are a group of abietane-type norditerpenoid quinones, originally found in danshen [62]. More than 40 structurally diverse tanshinones have been isolated and identified [67], among which cryptotanshinone, tanshinone IIA, and tanshinone I are the main active ingredients [68]. Although many biotechnological improvements have been implemented to increase tanshinone production from plants, at present no mature hairy root, suspension cell line, or culture system of *S. miltiorrhiza* have been developed. Thus, the extraction from roots and rhizomes of *S. miltiorrhiza* still represents the main source of tanshinones [62]. *Salvia yangii* has also been found to produce a range of tanshinones [69–72], as well as *S. abrotanoides*, although the compound assortment was found to be considerably different according to the preliminary data obtained by our working group.

Tanshinone I and tanshinone IIA display a variety of biological activities [39]. Tanshinone I is reported to induce apoptosis in leukemia cells [73], human colon cancer cells [74] and activated hepatic stellate cells [75], and displays anticancer effects in human non-small cell lung cancer [76] and human breast cancer [77]. Tanshinone IIA exerts a cardiovascular action [78], including effects against cardiomyocyte hypertrophy [79], atherosclerosis [80], hypertension [81] and ischaemic heart diseases [82]. In addition, tanshinone IIA is a potent anticarcinogenic, with possible application for the management of systemic malignancies [83].

As introduced above, tanshinone IIA is currently in short supply because of overcollection of the wild plants and environmental change [28], so that endophytic fungal strains represent an alternative source. In this respect, tanshinone I and tanshinone IIA production has been confirmed by *T. atroviride* D16 from *S. miltiorrhiza* [39]. Moreover, strain TR21 of *Aspergillus foeniculicola* was shown to produce low amount of tanshinone IIA [84]. Production of this compound by TR21 was increased in the NU152 mutant, obtained by traditional mutagenesis using ultraviolet radiation and sodium nitrate

treatment [28], and in strain F-3.4 through genome shuffling [85], providing a yield of tanshinone IIA which is over 11 times higher than the original strain TR21. This study showed that the genetic basis of high-yield strains can be achieved through genome shuffling, which can shorten the breeding cycle and improve the mutagenesis efficiency in obtaining strains with good traits, to be used for industrial production.

Cryptotanshinone, another nor-abietanoid diterpenoid, which is a main bioactive compound of *S. abrotanoides* known for leishmanicidal, antiplasmodial and cytotoxic activity [86] has been found to be produced also in roots of *S. yangii* [69]. Very recently, this compound has been reported as a secondary metabolite of endophytic strains of *P. canescens*, *Penicillium murcianum*, *Paraphoma radicina*, and *Coniolariella hispanica*, independently of the host plant. Moreover, the effect of exogenous gibberellin (GA3) on *S. abrotanoides* and endophytic fungi was shown to have a positive effect on increasing the cryptotanshinone production in the plant as well as in endophytic fungi cultivated under axenic conditions [22]. Exogenous gibberellin treatment was also previously observed to promote the production of cryptotanshinone, tanshinone I and tanshinone II in *S. miltiorrhiza* [87].

The typical abietane diterpenoid, ferruginol, is mainly known from *Sequoia sempervirens* for its antibacterial and antineoplastic properties [88,89]. It has also been isolated from the roots of plants in the genus *Salvia*, for instance *Salvia viridis* [90], *S. miltiorrhiza* [91], *Salvia cilicica* [92], *Salvia deserta* [93]. As a precursor in the tanshinone pathway, ferruginol synthesis has been confirmed by the above-mentioned strain D16 of *T. atroviride* [39].

*5.2. Endophytic Fungi as Biotic Elicitors*

Indiscriminate collection and cutting down of medicinal plants from the wild for extraction of medicinal products have almost led to the extinction of certain plant species, making them either vulnerable or critically endangered. The biotechnological approaches involving plant cell, organ and hairy root cultures appeared to fulfill the ever-increasing demand up to a certain level [54]. Endophytes could possibly be used as alternative or more efficient elicitors, compared to other biotic and abiotic elicitation methods.

A tanshinone IIA-producing endophytic strain of *A. foeniculicola* (U104) was demonstrated to elicit production of this compound in sterile seedlings of *S. miltiorrhiza* through upregulation of several enzymes involved in its biosynthesis [94]. Likewise, mycelium extract and its polysaccharide fraction (PF) produced by *T. atroviride* D16 promoted root growth and stimulated the biosynthesis of tanshinones in hairy roots. Moreover, the transcriptional activity of genes involved in the tanshinone biosynthetic pathway increased significantly after treatment with PF, which could be effectively utilized for large-scale production of tanshinones in the *S. miltiorrhiza* hairy root culture system [95]. Later on, PF was found to more deeply regulate the metabolic profiling of roots of this plant [96]. The main component of PF resulted to be an heteropolysaccharide (PSF-W-1), whose structure has been elucidated [97]. Moreover, an enhancing role by jasmonic acid on production of tanshinone I by this fungal strain was demonstrated [26], along with $Ca^{2+}$ triggering, peroxide reaction and protein phosphorylation, leading to an increase in leucine-rich repeat (LRR) protein synthesis [98].

Another endophytic strain from *S. miltiorrhiza* (*Phoma herbarum* D603) was found to stimulate growth and root development by producing IAA and siderophores and improving nutrition through phosphorus solubilization; moreover, it promoted the synthesis and accumulation of tanshinones by regulating the expression level of key genes in the synthetic pathway [37].

Eliciting effects on the synthesis of salvianolic acids and tanshinones, particularly dihydrotanshinone I and cryptotanshinone, have been also reported by a strain of *Chaetomium globosum* and its mycelial extract [29]. The effect of the mycelial extract was much stronger than that of the live fungus on tanshinones synthesis, which significantly increased the transcriptional activity of key genes in tanshinone biosynthetic pathway. Thus, *C. globosum* D38 was proposed to be supplemented as a biotic fertilizer in *S. miltiorrhiza* seedling culture, as it not only significantly promoted growth of the host plant, but also notably enhanced the accumulation of tanshinones and salvianolic acids.

*Alternaria* sp. A13 has been shown to simultaneously enhance the dry root biomass and secondary metabolite accumulation of *S. miltiorrhiza*, thus demonstrating its application potential as a bio-fertilizer in the cultivation of this plant [26]. Compared to uninoculated seedlings, *S. miltiorrhiza* seedlings colonized by *Alternaria* sp. A13 showed significant increment in the contents of total phenolic acids and lithospermic acids A and B. Examination of the related enzyme activities showed that the elicitation effect of A13 on lithospermic acid B accumulation correlated with cinnamic acid 4-hydroxylase (C4H) activity in the phenylpropanoid pathway under field conditions. A similar effect was demonstrated for a strain of *Paecilomyces* sp. which increased content of salvianolic acid B in *S. miltiorrhiza* and promoted plant growth [36].

### 5.3. Biotransformation/Detoxication Abilities of Endophytic Fungi

To be able to colonize host tissues, endophytes developed a strong tolerance toward host's defensive metabolites. The detoxification of plant bioactive compounds is an important transformation ability of many endophytes which, to a certain extent, decides the colonization range of their hosts [17]. Biotransformation abilities of endophytes help in detoxification of antifungal metabolites produced by the host plant, and may intervene in the production of some novel bioactive compounds [54,99,100].

*Trichoderma hamatum*, an endophytic fungus inhabiting roots of *Salvia officinalis* alongside other microorganisms, was found to be able to degrade caffeine [40]. Aromatic plants such as sage have been used as intercrops in coffee plantations. *Salvia officinalis* was proved to absorb caffeine from the incubation media and store it mainly in roots. The cited study demonstrated that the degradation of caffeine was initiated by the ability of the microorganisms to perform demethylations, whereas xanthine degradation may be attributed to either the plant or the microorganisms. The existence of a beneficial biochemical interaction in caffeine degradation between endophytic *T. hamatum* and sage root was proposed. Using sage with its endophyte *T. hamatum* as an intercrop may become an ecologically friendly strategy to reduce caffeine accumulation in soil.

## 6. Conclusions

Endophytic fungi are prospective producers of both known and novel bioactive compounds. However, to ensure feasibility of industrial application, yield and productivity enhancement strategies at several levels are required [101]. A combination of genetic, metabolic and bioprocess engineering may be used to sustain and enhance production of high value secondary metabolites from selected strains, whose biosynthetic abilities can be improved through physical and chemical mutagenesis, or various methods for genetic transformation. Improved strains can be in turn subjected to various bioprocess optimization strategies for further enhancement in yield and productivity of selected compounds.

This review of the available literature specifically concerning endophytic fungi of sages highlighted that research in the field is quickly progressing, with the aim of both refining biotechnological applications concerning tanshinone production and prospecting novel strains for further applications. The spread of reliable methods for detection and characterization of both the endophytic strains and their bioactive secondary metabolites is expected to further improve the translational perspectives.

**Author Contributions:** Conceptualization, B.Z.; investigation, B.Z.; resources, M.B., B.Z.; writing—original draft preparation, B.Z., M.B., B.A., R.N.; writing—review and editing, M.B., R.N.; funding acquisition, B.Z., M.B. All authors have read and agreed to the published version of the manuscript.

**Funding:** This research received no external funding.

**Conflicts of Interest:** The authors declare no conflict of interest.

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
