# Peer review of "Bioactive Products from Endophytic Fungi of Sages (Salvia spp.)"

_agriculture, doi:10.3390/agriculture10110543_

Round 1

Reviewer 1 Report

This is a very well written review on the upcoming subject of extraoridnary properties of endophytic fungi. The review takes into consideration all aspects related to the subject, extensively listing the bioactive secondary metabolites discovered to date as well as the biotechnological implications of their use. The excellent use of the English language enables the understanding of the extended details presented. Two typing mistakes follow:

  • 42: Lamiaceae “is” one of the most important families
  • 91: “… to such an extent that… “. Please change, no capital T.

Author Response

Dear Reviewer,

Thank you very much for the detailed revision of our manuscript. We truly appreciate your positive feedback. We also do appreciate the time and effort spend to revise our work. We have made the amendments in the text to correct the typing mistakes, which are highlighted in red in the revised version of the manuscript.

We do hope that this revised manuscript is now appropriate for publishing in Agriculture.

Best regards,

Monika Bielecka & Co-authors

Reviewer 2 Report

Dear Authors,

I think your paper needs incorporation of some revisions.

The coverage of the literature is quite good, but not complete. The scopus search "TITLE-ABS-KEY((salvia OR sage) AND (endophy* AND (isol* OR purif*)) AND fung*)" resulted in 43 documents at the time of writing. The results of the following relevant articles were were not included in your paper: 10.5428/pcar20130310; 10.5428/pcar20160612; 10.1080/14786419.2019.1616727. Please address the issue by running similar searches for all other included plant genera.

I understand that Table 2 contains those bioactivities only that were tested in the cited studies, but many if not most of the compounds have described bioactivities beyond those listed (most notable examples may include caffeic acid, daidzein). Please add additional data on these compounds (possibly in a supplementary material, or, interesting ones might be highlighted in the main text).

Some parts of the paper (5.2., perhaps also 5.3.) do not belong to the topic addressed by the title. A more focused approach should be used, you should consider removing the mentioned sections. As an alternative, consider a broader scope in the title.

Fig.1.: Is this own work or photo from a different publication? Either way, add reference(s). If it is your unpublished work, add details on methods in a supplementary material. The same applies to Table 1 rows with the remark "this paper".

There is a taxonomic name typo in Fig.2. x axis: correct to S. miltiorrhiza.

Rather than showing all compounds in a simple alphabetic order, some classification should be incorporated into Table 2.

Additional notes should be added where the endophytic fungus was shown to contain the same compound as the host plant, as this has shown to be an artefact (storage of residual compounds) in some instances, please see 10.1007/s13225-013-0228-7 for an example (taxol). Non-polar compounds could also be subject to similar phenomena.

Best regards.

Author Response

Dear Reviewer,

Thank you very much for the detailed revision and corrections proposed to improve our manuscript. We do appreciate the time and effort spend to improve our work. We did our best to suitably address all issues raised by performing appropriate changes into the manuscript and provided extensive explanations in the correspondence below. All changes in the revised manuscript are marked in red.

Actually, we had preliminarily quite carefully checked the pertinent literature available from Google Scholar, including the genera which have been integrated in Salvia. In this respect, papers referring to the doi 10.5428/pcar20130310 and 10.5428/pcar20160612 are not available as full texts, while the recently published paper 10.1080/14786419.2019.1616727 is relevant and we integrated information in the text, Figure 2 and tables.

As for Table 2, we deliberately decided not to consider information inherent bioactivities reported in miscellaneous literature for all compounds because this was outside our scopes and the chosen domain of investigation. Conversely, sections 5.2 and 5.3 are related to the issue of bioactive compounds from endophytic fungi, at least with reference to the interactive aspects with the host plant and the possible biotechnological exploitation.

Figure 1 is original, and reference has been added in caption.

For Table 2 we also prefer not to bring any adjustment, considering that alphabetical order can be helpful in way of quickly finding compounds of particular interest by the reader.

As for the final comment that reports concerning the finding of plant products in cultures of endophytic strains are artefact, we absolutely dissent from this position. Actually, paper by Heinig et al. has remained quite an isolate assertion which has been overcome by a multitude of observations that indeed, both for taxol and othe bioactive products, endophytic fungi share these biosynthetic abilities with the host plants.

We do hope we have suitably addressed all comments and that this revised manuscript is now appropriate for publishing in Agriculture.

Best regards,

Monika Bielecka & Co-authors